# Feasibility of Non-Invasive Sentinel Lymph Node Identification in Early-Stage NSCLC Through Ultrasound Guided Intra-Tumoral Injection of ^99m^Tc-Nanocolloid and Iodinated Contrast Agent During Navigation Bronchoscopy

**DOI:** 10.3390/cancers16223868

**Published:** 2024-11-19

**Authors:** Desi K. M. ter Woerds, Roel L. J. Verhoeven, Erik H. J. G. Aarntzen, Erik H. F. M. van der Heijden

**Affiliations:** 1Department of Pulmonary Diseases, Radboud University Medical Center, 6525 GA Nijmegen, The Netherlands; desi.terwoerds@radboudumc.nl (D.K.M.t.W.); erik.aarntzen@radboudumc.nl (E.H.J.G.A.); erik.vanderheijden@radboudumc.nl (E.H.F.M.v.d.H.); 2Department Nuclear Medicine and Molecular Imaging, University Medical Center Groningen, 9713 GZ Groningen, The Netherlands; 3Department of Nuclear Medicine, Eberhard Karls University Tuebingen, 72074 Tuebingen, Germany

**Keywords:** non-small cell lung cancer, navigation bronchoscopy, radial ultrasound, sentinel lymph nodes, imaging tracer, fluoroscopy, SPECT imaging, SLN procedure

## Abstract

This clinical feasibility study explored the use of a novel device that integrates radial ultrasound and an angulated needle to inject ^99m^Tc-nanocolloid and CT contrast for visualization of the sentinel lymph nodes (SLN) using SPECT and CBCT imaging. Radial ultrasound could visualize 29 out of 30 tumors. Injection of imaging tracers was feasible in all patients. Injection with CT contrast under fluoroscopy guidance could predict the dissipation of ^99m^Tc-nanocolloid and facilitate the repositioning of the needle, when necessary. SPECT imaging after ^99m^Tc nanocolloid injection seems insufficiently sensitive, allowing for detection of an SLN in only 32% of cases.

## 1. Introduction

While advancements in medicine have improved cancer treatments, overall lung cancer survival is still abysmal. Moving towards early-stage disease detection and treatment optimization are considered key strategies to tackle non-small cell lung cancer (NSCLC) [1].

In the early stages of NSCLC, it is paramount to determine the extent of disease, as this has profound implications for the survival of patients. While the size of the primary tumor impacts survival, i.e., 5-year survival for stage IA (10–30 mm) disease is 88% and for stage IB (30–40 mm), it is 78%, lymph node involvement has an even more detrimental effect. The 5-year survival for patients with stage IA disease (88%) decreases to 68% as soon as an N1-node is involved and to 50–61% when N2-disease is discovered [2]. Unfortunately, current lymph node staging strategies are suboptimal. We and others have shown that guideline-concordant staging via the combined availability of positron emission tomography/computed tomography (PET/CT) imaging, systematic endobronchial ultrasound (EBUS), and/or mediastinoscopy is accurate in only 65.5% of resectable early-stage (cN0) NSCLC. Occult lymph node metastases were found in 9% to 23.1% of patients after surgery with curative intent [3,4,5].

It is known that lymphatic spread often starts in lymph nodes which are beyond the reach of EBUS and thus cannot be assessed during diagnostic work-up. These lymph nodes, which are most likely to harbor lymphatic spread, are called the sentinel lymph nodes (SLN), and they should be accurately identified [6]. Imaging procedures to identify the SLN are routinely implemented in other tumor types, such as breast and skin cancer [7,8]. During an SLN procedure, several intra- and/or peritumoral injections of a tracer with radioactive, fluorescent, or magnetic properties are administered, which drains via the lymphatic system and accumulates in lymph nodes over time. This tracer can be visualized before or during surgery to guide SLN identification [9].

With the advent of neoadjuvant drug therapy in early-stage lung cancer, as well as knowing that only surgery would allow for resection for further exploration of the nodes, a universal minimally invasive approach is preferred. An approach that allows for the identification of such nodes on non-invasive imaging such as single-photon emission computed tomography (SPECT), PET, or CT imaging prior to therapy could prove beneficial, particularly in early-stage disease. Additionally, given the further development of image-guided navigation bronchoscopy (NB) procedures in recent years, minimally invasive access to small primary lung cancers has improved [10,11,12]. Ideally, the SLN would be visualized on imaging immediately following an NB for diagnostic purposes, which may allow for a better patient selection for local treatments.

To integrate an SLN procedure into an NB procedure requires specific endobronchial instruments. The placement of an injection should preferably be performed when the operator is confident that the tracer is injected into the lesion or its direct surroundings. As larger lesions can drain to a greater area, repeat local repositioning of the injection instrument into different areas of the tumor or its surrounding tissue should be feasible [13]. Additionally, injections should preferably be guided by real-time 3D imaging to allow for understanding of the positioning of instruments relative to the tumor [14,15,16]. At present, commonly used devices for bronchoscopy do not meet all these requirements. An ultrathin bronchoscope can visualize an additional 3–4 bronchial generations beyond those captured by a conventional bronchoscope, but some lesions might still be unreachable, the camera can be obscured by mucus or (local) bleeding, or the bronchoscope could become wedged. Radial EBUS (rEBUS) is widely available and is a valuable tool for local position confirmation, as it can visualize structures beyond the bronchial wall, but it does not provide directional support for real-time tissue acquisition or injection guidance like linear EBUS-systems [16,17].

In this single-center explorative study, we evaluated the feasibility of adding an SLN procedure to NB by introducing a novel instrument that combines radial ultrasound (US) and a retractable angulated needle that enables intra- and peritumoral injections under real-time imaging. We evaluated (1) the possibility of injecting a radiotracer and iodinated contrast agent in peripheral pulmonary lesions using this instrument, and (2) whether these tracers allow for subsequent identification of drainage to the SLN on SPECT/CT imaging.

## 2. Materials and Methods

### 2.1. Study Design and Patient Cohort

This is a prospective, single-center clinical feasibility study of an endobronchial SLN procedure following a diagnostic NB procedure in patients suspected of having early-stage lung cancer. Patients that were referred for a diagnostic navigation bronchoscopy to diagnose a lung lesion of 10 to 50 mm, with or without ipsilateral hilar node involvement (stage IA-IIB), were eligible. Patients aged < 18 years, with an ASA score > 4, previous treatment for lung cancer, or a known allergy to contrast agents were excluded. Pre-procedural staging was based on combined fluorodeoxyglucose ([^18^F]FDG-)PET/CT or contrast chest CT imaging alone. Patients with imaging-based suspicion of N2 or N3 lymph node metastases were excluded. Patients were enrolled between December 2022 and May 2024 after providing informed consent. This trial was approved by the Local Medical Research Ethics Committee (Reference No. 2022-13640 on 29 November 2022) and followed the ethical principles of the Declaration of Helsinki (clinicaltrials.gov identifier NCT05555199) [18].

### 2.2. Navigation Bronchoscopy Procedure

All patients underwent a diagnostic NB in which a therapeutic bronchoscope (19-J10 endoscope, Pentax Medical, Tokyo, Japan) and a pre-angulated extended working channel of 180 degrees or medial curvature (Medtronic EWC, Minneapolis, MN, USA) were navigated to the lesion under cone beam computed tomography (CBCT-, Philips Azurion Flexarm or Siemens Artis Pheno), augmented fluoroscopy (AF-), and radial endobronchial ultrasound (rEBUS) guidance (Olympus, Tokyo, Japan), as reported previously [17,19]. Biopsies of the lesion for tissue diagnosis were taken by TBNA and/or forceps and were assessed by rapid on-site evaluation (ROSE), as per clinical practice. When no benign diagnosis was found by ROSE, and the lesion was still suspected of lung cancer, the patient was further enrolled in the study. If CBCT imaging after biopsy showed any grade of blood in the surrounding lung parenchyma prior to imaging tracer injection, this was noted. Finally, a systematic EBUS was performed to assess the involvement of hilar and mediastinal lymph nodes, following routine clinical practice (EB-19-J10U, Pentax Medical, Japan).

### 2.3. Study Procedure

#### 2.3.1. Pioneer Plus Radial Ultrasound

Prior to this study, an explorative ex vivo study was performed to test the feasibility of using the intravascular ultrasound (IVUS) Pioneer Plus catheter (Philips IGT-D, Amsterdam, The Netherlands) for radial US-guided injection in resected human lung tissue with ^99m^Tc-ICG-nanocolloid [20,21]. Having evaluated the feasibility of injection via varying volumes and imaging parameters, the Pioneer Plus was used outside of its intended use, with permission of the medical ethics committee, for evaluation in this clinical feasibility study.

The Pioneer Plus was originally designed as an intravascular catheter intended to identify and re-enter the true vessel lumen based on IVUS imaging. It does so by using a 20 MHz ultrasound transducer with solid-state piezo-electric crystals, allowing a maximum imaging diameter of 20 mm. The Pioneer Plus further includes a nitinol-based needle that, upon extension outside of the sheath, comes out from the catheter at a 90° angle at a maximum needle extension of 7 mm (see Figure A1). The catheter is torsionally stiff and can thus be rotated for adequate needle (re)positioning. The ultrasound probe is calibrated into the catheter such that the angulated needle does not come into the direct imaging plane of the ultrasound imaging upon full extension but stays just proximal of the transducer (~2 mm). The imaging plane is perpendicular to the needle extension, and the needle will always extend out of the sheath at the 12 o’clock angle in the ultrasound image. This calibration allows for validation that the catheter is correctly positioned relative to the tumor such that adequate injection can take place under real-time imaging.

#### 2.3.2. Imaging Tracer Injection

An intention to inject multiple tracer depots was established based on pre-procedural planning and intra-procedural findings. To perform the injections, the working volume of the Pioneer Plus catheter of 0.3 mL [20] was pre-filled using a 3 mL Luer-lock syringe filled with 2 mL ^99m^Tc-nanocolloid solution. The Pioneer Plus was inserted into the pre-angulated catheter that remained positioned near or inside the lesion. Using the IVUS-console (Core M2, Philips, The Netherlands), the lesion and injections could be imaged using radial US. Upon verifying the positioning of the catheter and lesion on continuous US imaging and/or (augmented) fluoroscopy (Figure 1), the target location is fixated at the 12 o’clock position. Injection(s) were placed either in the tumor—intratumorally—or surrounding the tumor—peritumorally. Both approaches were performed to assess feasibility, also depending on the feasible positions of the catheter, as concluded during the navigation bronchoscopy procedure. The 7 mm (angulated) reach of the needle was used in all injections to ensure sufficient penetration beyond the bronchial wall. After the first injection, the needle was retracted, and the catheter was navigated to new locations near or in the lesion to place multiple tracer depots, when possible. After the last injection, the needle was retracted, and all radioactive tracer in the working volume was removed by placing a vacuum on the syringe proximally.

Using this approach, an attempt to evenly administrate the tracer in or around the lesion in multiple depots was made using visualization of the lesion on radial US and fluoroscopy [22]. The injected volume and number of injections was varied to study the characteristics for a successful injection, but tumor volume was taken into account to prevent injecting more than 50% of the total tumor volume.

#### 2.3.3. Protocol Amendment

At the start of the study, all subjects underwent at least one injection of ^99m^Tc-nanocolloid (Nanocoll, GE Healthcare, Eindhoven, The Netherlands). Interim analysis after the treatment of 10 patients showed that the SLN could not always be visualized on SPECT/CT following the injections (6/10 patients). It was thereby concluded that immediate, real-time visualization to gain feedback on tracer dissipation may improve injection accuracy. Subsequently, an iodinated contrast agent for CT imaging, Iomeron 300 (300 mg iodine/mL, Bracco Imaging Deutschland GmbH, Konstanz, Germany), was added to the injection protocol (with ethical committee approval) from subject 15 onwards. When, despite US guidance during injection, fluoroscopy showed endobronchial or direct vascular leakage of Iomeron 300, the position of the needle could be adjusted prior to ^99m^Tc-nanocolloid injection.

#### 2.3.4. SPECT/CT Imaging and SLN Identification

Initially, two SPECT/CT scans were performed within a window of 1 to 6 h after injection to allow for the evaluation of appropriate scan time frame(s) (for imaging parameters, see Appendix A). During the interim analysis, the optimal timing of SPECT/CT scanning was evaluated, whereafter all other patients received one SPECT/CT scan at an optimal interval. The lymphatic drainage and uptake of ^99m^Tc-nanocolloid and Iomeron in the lymph nodes as observed via SPECT/CT imaging was assessed by a nuclear medicine physician (EHJGA).

### 2.4. Data Analysis

Patient, lesion, and injection characteristics; fluoroscopy-, CBCT-, radial US- SPECT/CT imaging; and pathological outcome were collected of all patients. This was an explorative feasibility study, and only descriptive statistics are described.

## 3. Results

A total of 34 patients were included in the study. Three patients were excluded due to a lack of intra-operative confirmation or suspicion of malignancy by ROSE. Thirty-one patients with a median lesion size of 18.7 mm underwent the study procedure following successful NB. The patient and lesion characteristics are depicted in Table 1. The study procedure added around 15 min to the procedure time, making the total procedure time a median of 1 h 45 min; see Table 2.

### 3.1. Pioneer Plus Radial Ultrasound

The Pioneer Plus was used to perform tracer injection(s) in 30 out of 31 patients (Table 2). A conventional aspiration needle was used once when a non-solid lesion was located directly at the tip of the extended working channel, and the angulated needle of the Pioneer Plus catheter could only achieve a peritumoral injection (where intratumoral injections are preferred in non-solid lesions [20]). Radial US imaging could visualize 29 out of 30 lesions (96.7%). One lesion of 20.6 mm in the LUL was not visible on radial US imaging due to loss of contact with the lesion because of deformation of the extended working channel upon insertion of the relatively rigid Pioneer Plus catheter through its lumen, which rendered the US imaging unusable for guiding the injection (fluoroscopy/CBCT imaging was used instead). Figure 2 visualizes the radial US imaging of different lesion types. Unfortunately, none of the injections itself were visible on radial US, as the needle extension did not result in in-plane visualization, as known from linear-EBUS, and injected tracers were echolucent.

Figure A1 and Figure A2 indicate the visibility of the radial US Pioneer Plus on the fluoroscopy and CBCT images.

### 3.2. Imaging Tracer Injection

#### 3.2.1. ^99m^Tc-Nanocolloid Injection

A median of two injections (range, 1–4) with a volume of 0.4 mL (range, 0.2–1.0 mL) and a Technetium-99m activity dose of 43.9 MBq (range, 8.1–51.6 MBq) were administered in 31 patients. Placement of multiple depots was performed in 22 out of 31 patients (73.3%). The other nine patients received an injection of imaging tracer(s) at a single location, as multiple depot placement was not considered to be of additional value due to catheter placement next to the lesion (*n* = 2) or previous intra-parenchymal bleeding from diagnostic biopsy hampering the clear distinction of the lesion from the parenchyma (*n* = 7).

#### 3.2.2. Iodinated Contrast Injection

A median of one Iomeron 300 injection (range, 1–4) with a volume of 0.3 mL (range, 0.3–1.0 mL) was performed in 15 patients. Iomeron visualized leakage to a vessel on fluoroscopy that was not visible on radial US imaging in 4 out of 15 patients (26.7%). This enabled the relocation of the needle to a different location to ensure an accurate intra- or peritumoral injection of ^99m^Tc-nanocolloid for drainage to the lymph nodes. As can be seen in Figure A3 and Figure A4, the injected depot of Iomeron 300 can be visualized on CBCT and subsequent low-dose CT imaging.

### 3.3. SPECT/CT Imaging and SLN Identification

In the first ten patients, two SPECT/CT scans were performed (see also Figure A5 for a flowchart of patients and study procedures). Early scans were performed between 1 and 3 h after injection, and a secondary late scan was performed between 3 and 6 h after injection. Example images are given in Figure 3. After ten patients underwent two SPECT/CT scans following ^99m^Tc-nanocolloid injection, an interim analysis was performed to determine the need for two time frames. Only one patient had an SLN identified in the late scan (at 04:44 h). which was not visible in the early scan. It was therefore concluded that the optimal scanning time frame was after at least 4 h, and that one scan would suffice (exact timing was always co-dependent on scanner availability and logistics).

In total, an SLN was identified on SPECT/CT imaging (early as well as late scan times) in 10 out of 31 patients (32.3%); see Table 3. When an SLN was identified on SPECT imaging, it was only one lymph node in seven patients (22.6%) and two lymph nodes in three patients (9.7%).

As an SLN could be identified on SPECT/CT in only a minority of patients, the imaging acquisition time was prolonged by 120% (from 10 min to 22 min) in the last four patients to check whether this would increase sensitivity. An SLN was identified in only one of these patients (25%).

Leakage of ^99m^Tc-nanocolloid itself could not be visualized on fluoroscopy, CBCT, or radial US imaging. However, scattering of the radioactivity of ^99m^Tc-nanocolloid throughout the lung or segment was visible in nine patients (36.0%) on SPECT/CT, suggesting endobronchial leakage.

### 3.4. Lesion and Lymph Node Outcomes

One patient was diagnosed with station 4L involvement through same-procedure EBUS post-navigation bronchoscopy (but did not show an SLN on SPECT imaging).

As per multidisciplinary team decision, fifteen patients underwent surgical resection, allowing for histopathological confirmation. Post-operative detailed assessment of lymph node involvement was found in three patients (Table 3). One patient was found to have macro-metastatic disease in stations 3 and 5 (pN2), and two patients were diagnosed with isolated tumor cells (ICTs). In none of these three patients was an SLN found on SPECT/CT imaging, making determination of a true SLN impossible.

### 3.5. Safety, Risks, and Adverse Events

The study procedure involved additional radiation by means of fluoroscopy-guided needle placement and injection, which comprised approximately one to two minutes. A maximum of two extra CBCT scans were additionally made to visualize tool-in-lesion and the dissipation of Iomeron in and around the lesion in the study setting. The radiation dose and safety of CBCT-guided NB has been previously described by Verhoeven et al. (2021) and Wijma et al. (2024) [24,25]. The injection of ^99m^Tc-nanocolloid with a maximum dose of 100 MBq adds ~0.5 mSv to the procedure but is predominantly localized near the lesion itself, resulting in a minimal increase in total effective dose per patient. After NB, up to two low-dose CT scans of around 3.3 mSv each are performed to provide anatomic reference to the detected ^99m^Tc.

One serious adverse event was observed. Upon symptoms, a patient returned two days after the procedure with a pneumonia in the biopsied area. In this specific patient, a Nashville grade 2 bleeding had occurred during NB, for which adrenaline was successfully locally applied through the catheter prior to the study procedure using ^99m^Tc-nanocolloid injections [26]. It is therefore thought to be caused by the biopsies taken before the study procedure and not by the use of the Pioneer Plus catheter or the placing of the imaging tracer depots.

## 4. Discussion

This single-center explorative study shows that an SLN procedure can be performed following a diagnostic NB procedure through use of a novel multi-modal catheter that allows for the accurate intra-/peritumoral injection of tracers under US and fluoroscopic guidance. The multi-modal Pioneer Plus catheter was able to visualize all tumors and allowed for the injection of imaging tracers in all cases. The combination of secondary imaging techniques, such as using iodinated contrast, fluoroscopy, and CBCT imaging, further allowed for the evaluation of tracer leakage beyond the ultrasound plane. While the injection of intra-/peritumoral tracers was feasible in all cases, subsequent drainage of tracers into an SLN could not be uniformly identified on SPECT/CT imaging. ^99m^Tc-nanocolloid and/or SPECT imaging as an imaging tracer might not provide suitable ingredients for a non-invasive SLN procedure in lung cancer when employed immediately following diagnostic navigation bronchoscopy.

### 4.1. Real-Time US-Guided Endobronchial Injection

The Pioneer Plus catheter, originally designed and approved as a vascular lumen re-entry catheter, proved to be an effective and safe device that could be used for the endobronchial injection of radiotracers, as used in this study. During the design and conduction of this trial, there were no other devices available that combined US and a needle for real-time injection. However, as of 2023, the iNod Ultrasound Guided Biopsy Needle (Boston Scientific, Marlborough, MA, USA) became available in the USA for image-guided biopsy, although it is not (yet) available in the EU [27]. The iNod is similar to the Pioneer Plus catheter but exhibits some differences in regards to its technical aspects. While the Pioneer Plus has an angulated 24 G needle that stops just before the US field of view, the iNod has a 25 G needle that exits at an 11° needle ramp and is within the US field of view. The iNod uses a single-element rotational transducer with a frequency of 30 MHz, classifying it as an ultra-high frequency device, whereas the Pioneer Plus employs a 20 MHz transducer with continuous radial ultrasound generated by 64 fixed elements [28]. Due to its lower frequency and continuous imaging at all angles, the Philips system potentially provides better overall image quality and a larger imaging diameter, which can potentially improve the view of the lesions further away from the bronchi when used endobronchially.

The Pioneer Plus catheter has been demonstrated to be safe and effective in injecting imaging tracers in or around peripheral lung lesions. The combination of radial ultrasound and a needle also makes this type of device necessary when moving towards local treatments. Real-time visualization of the tumor by radial US, in combination with the 3D imaging possibilities of CBCT for precise positioning and confirmation, could provide the essential support for safely administering therapeutic agents locally.

### 4.2. Imaging Tracers and Other Particles

Over the past two decades, several research groups have studied the feasibility of an SLN procedure for lung cancer, with heterogenous approaches and results. These protocols have varied in several aspects, including whether the radiotracer was injected intraoperatively (one-day protocol) or transthoracically by CT guidance before surgery (two-day protocol). Additionally, differences are observed in the injected volume, activity, methods of injection, and time to detection. Various particles, namely ^99m^Tc-sultur-colloid (61–445 nm) or ^99m^Tc-tin-colloid (100–1000 nm), were used [29,30,31,32,33,34,35]. Kim et al. hypothesize that the ideal particle size for SLN mapping is in the range of 10–50 nm, suggesting that larger particles may be retained at the injection site, while smaller sizes are cleared from the injection site too quickly [36].

In our study, ^99m^Tc-nanocolloid (Nanocoll, GE Healthcare, The Netherlands) was used (particle size of 6.5–68 nm, with 99% < 13 nm), which could be considered to be within the ideal particle size range [9,37]. Despite this, while previous studies using gamma probe detection achieved SLN identification over 60%, our SLN identification rate on SPECT imaging by ^99m^Tc-nanocolloid remains at only 32.3% [29,30,31,32,33,34,35]. There are multiple factors that influence the clearance and retention of particles in the lymph nodes, e.g., coating, surface charge, colloidal stability, and biological compatibility. However, the ideal particle for an SLN procedure and the best method of detection remain unclear [38], as is further corroborated by our findings. Future research is needed to pinpoint the specific dynamics of individual tracers and carriers, which would allow for evidence-based protocols for SLN imaging identification.

### 4.3. Imaging Modalities

To the best of our knowledge, three studies have used a transthoracic injection of a radiotracer for SPECT imaging of sentinel lymph nodes in lung cancer patients. Romano et al. injected 10 mL of ^99m^Tc-tilmanocept intra- and peritumorally under CT guidance. They performed a SPECT scan in six patients after a median of four hours. Lymph node uptake was detected in four patients (66.7%) [39]. In addition, they used a gamma probe intraoperatively and detected an SLN in all patients. Nomori et al. performed a similar procedure and found an SLN in 39 out of 63 patients (62%) on SPECT imaging around 16 h after administration. They also used a gamma probe during surgery and detected an SLN in 48 patients (78%) [33]. Lastly, Abele et al. performed SPECT imaging 1, 2, or 3 h after CT-guided injection and found an SLN in 12 out of 24 included patients (50%) [40]. These groups found similarly low identification rates via SPECT imaging, and the two groups that included gamma probes identified more SLN intra-operatively than those observed via pre-operative SPECT imaging. This might be due to the SPECT spatial resolution and the long acquisition time of SPECT imaging, which can result in breathing artefacts, attenuation, and scattering of radioactivity by surrounding tissue and detector geometry. Therein, intraoperative gamma probes may offer advantages, as these can be located closest to the lymph node, significantly increasing sensitivity. Given that previous research on SLN identification using gamma probes reported significantly higher detection rates (61.5% to 86%) compared to our findings using SPECT imaging, more sensitive non-invasive imaging modalities should be considered for further exploration [29,30,31,32,33,34,35]. PET imaging, for example, offers superior resolution compared to SPECT and could potentially achieve higher detection rates for SLN, but there is currently no PET tracer available for this purpose. Additionally, the complexity of producing positron-emitting nuclides poses challenges for routine use in SLN procedures. To improve lymph node staging in the future, further exploration of alternative imaging modalities and protocols, in combination with optimized particle characteristics, is essential.

### 4.4. Strengths and Limitations

This study demonstrates that performing an SLN procedure in combination with an NB is technically feasible and can be performed through endobronchial access with a novel device that allows for real-time ultrasound visualization of the tumor, combined with simultaneous needle placement. However, there are still limitations that should be mentioned. This was a single-center study, making assessment of reproducibility difficult. We found that injection has consistently been shown to be feasible, but due to low overall SLN detection rates, the potential influence of injection volume on detection rates could not be investigated, as is the same for other injection and acquisition parameters in this exploratory study. Due to the low detection rate, and since we excluded patients with prior local or systemic cancer treatments, this study does not allow for conclusions to be drawn regarding the feasibility of SLN imaging in respect to (down-)staging after neo-adjuvant treatments.

After an interim analysis, we decided to check for injection dissipation by including iodinated contrast prior to ^99m^Tc-nanocolloid injection. With Iomeron 300, we were able to visualize the dissipation of the injected tracer and found leakage after first-attempt needle placement in 4 out of 15 patients, which was not expected based on real-time radial US imaging. Although the exact cause of leakage (bigger vessels, capillaries, or direct endobronchial leakage) was not clear in all patients, the location of Iomeron was always visible on fluoroscopy, even at low volumes (0.1–0.45 mL per injection).

We were successfully able to inject radiotracers in all patients. The attempted injection volume was based on predefined criteria ensuring that no more than 50% of the tumor volume was injected into a single lesion, as we hypothesized that higher volumes would increase the chance of endobronchial leakage. However, radioactivity was detected in other parts of the lung via SPECT imaging in nine patients (29.0%) throughout the study, which could be an indicator for intrabronchial leakage by post-procedural coughing. As the authors could not find studies that provided recommendations regarding volume, future studies of the optimal injection volume when related to lesion size and the chance of leakage should be performed.

To minimize the potential impact on patients but maximize the impact of sentinel node detection on clinical decision making, we choose to perform an SLN procedure at the end of a diagnostic NB, in contrast to previous studies reporting on SLN detection using intraoperative methods during surgery. Unfortunately, we could only detect an SLN in a minority of patients (32.2%), which led us to conclude that our current approach is insufficient. Lesion aspects like solidity and the presence or absence of cysts/cavities (25.8% in our cohort) or their combination with diagnostic sampling prior to tracer injection, which inherently results in minor (intra-parenchymal) bleeding (visible on CBCT in 38.7% of patients), are factors that may alter the local environment before attempting imaging tracer injection. However, our sample size is too small to explore such correlations. To prevent possible biopsy-related leakage, an SLN imaging protocol in which an NB procedure is performed solely to inject the (radio)tracer could increase the identification rate, but this would also significantly increase patient burden and the amount of effort required.

## 5. Conclusions

The injection of a (radio)tracer endobronchially using a device that combines radial US and an angulated needle is feasible, with an SLN identification on SPECT/CT imaging of 32.3%. The Pioneer Plus catheter was able to visualize all tumors and provided guidance while injecting imaging tracers. Iomeron could readily visualize the dissipation of the injected fluid in the lung. The obtained SLN identification by SPECT/CT imaging is lower than that obtained in other studies that focused on identification via more localized imaging (intra-operative gamma probe). Multiple parameters were varied and studied in this trial, but no protocol parameter could be optimized. Future research should be performed to find alternative, more sensitive, non-invasive SLN imaging techniques.

## Figures and Tables

**Figure 1 cancers-16-03868-f001:**
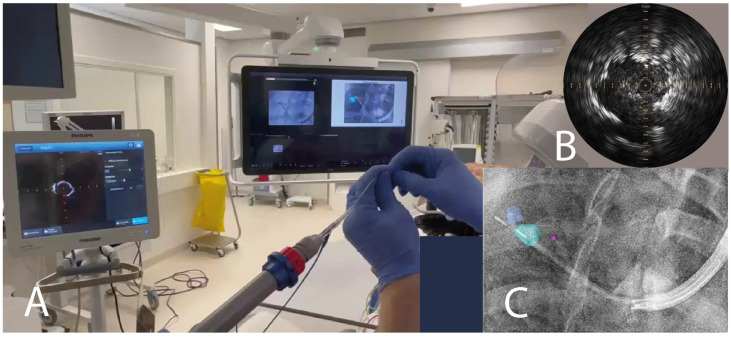
Visualization of the Pioneer Plus catheter in patient (**A**) with a view of the radial US image (**B**) and augmented fluoroscopy (**C**) in a suite equipped with a Philips Azurion Flexarm CBCT system. Augmented fluoroscopy depicts two parts of the lesion in dark and light blue and a pink dot for the catheter position on CBCT, used for navigation and sampling. The US transducer (distal) and needle shaft (proximal) can be seen as radiopaque in (**C**). Abbreviations: CBCT, cone beam computed tomography; US, ultrasound.

**Figure 2 cancers-16-03868-f002:**
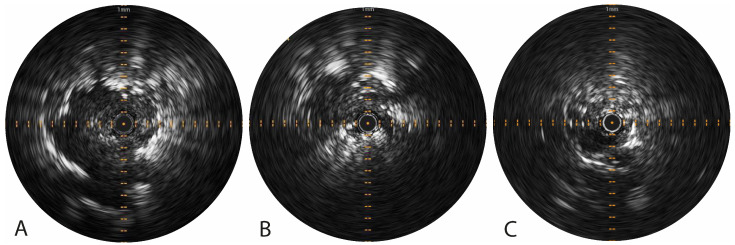
Visualization of a solid lesion (**A**), part-solid lesion (**B**), and GGO (**C**) on radial US imaging via the Pioneer Plus catheter after having completed diagnostic navigation bronchoscopy, pre-injection. (**A**) The solid lesion is clearly visualized. Beyond the lesion, some intra-parenchymal bleeding following previous sampling altered the echogenicity of the lung parenchyma. (**B**) The part-solid lesion is visualized, along with the GGO component that can be predominantly seen from 9 to 12 o’clock. Some hyper-echoic speckle is seen at the 4 to 7 o’clock position, following minor bleeding after biopsy. (**C**) The image of the GGO not only shows a mixed blizzard sign (as mentioned by Park et al. [23]) but also a clear (pulsating) vessel from the 6 to 9 o’clock position. The vessel should be out of the vicinity of the needle at the 12 o’clock position and should therefore stay in the 3 to 9 o’clock position during injection. Abbreviations: GGO, ground-glass opacity; US, ultrasound.

**Figure 3 cancers-16-03868-f003:**
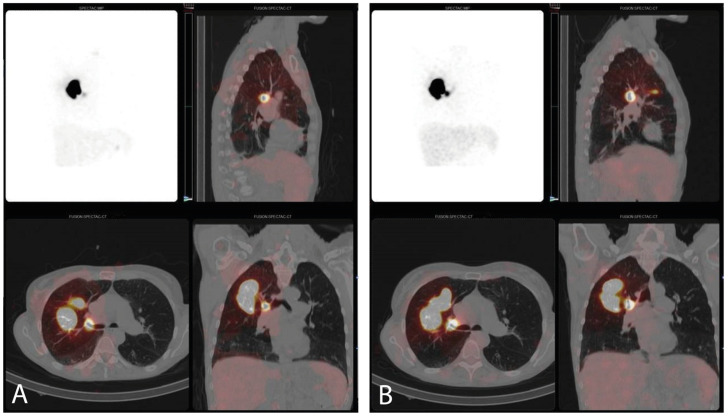
Visualization of the ^99m^Tc-nanocolloid detection in the coronal plane (upper left) and combined SPECT and low-dose CT images in the sagittal (upper right), transversal (lower left) and coronal plane (lower right) in one patient with an early scan time of 02:09 h (**A**) and a late scan time of 04:09 h (**B**). The SLN is visible in both scans, although more of the ^99m^Tc-nanocolloid seems to have drained to the lymph node in the late scan.

**Table 1 cancers-16-03868-t001:** Pre-procedural patient and lesion characteristics. ^1^ Cyst/cavity was scored positive when gas-containing structures were found within the tumor or its direct surroundings. ^2^ PET imaging showed slightly elevated FDG-avidity in nodal regions 5/6 in two cases. Abbreviations: BMI, body mass index; CT, computed tomography; DLCO, diffusing capacity of the lungs for carbon monoxide; FEV1, forced expiratory volume in the first second; GGO, ground-glass opacity; IQR, interquartile range.

	Characteristic		Frequency
Patient characteristics	Age, median (±IQR)		71 (±14)
	Gender, *n* (%)	Male	21 (67.7%)
		Female	10 (32.3%)
	BMI, median (±IQR)		26 (±5)
	FEV1, median (±IQR)		88 (±31)
	DLCO, median (±IQR)		84 (±23)
Lesion characteristics	Lesion size on CT, median (±IQR)		18.7 (±9.6)
	Lobe, *n* (%)	Upper	25 (80.6%)
		Lower	6 (19.4%)
	Lesion type, *n* (%)	Solid	19 (61.3%)
		Part-Solid	4 (12.9%)
		GGO	7 (22.6%)
		Cystic	1 (3.2%)
	Cyst/cavity, *n* (%) ^1^		5 (16.1%)
	Imaging-based pre-procedural stage, *n* (%)	iN0	24 (77.4%)
		iN0-1	5 (16.1%)
		iN1-2 ^2^	2 (6.5%)

**Table 2 cancers-16-03868-t002:** Study characteristics regarding the use of the Pioneer Plus catheter, ^99m^Tc-nanocolloid, and Iomeron 300 injection. ^3^ Bleeding after biopsies was scored positive when intra-parenchymal bleeding was visible on CBCT imaging and was furthermore noted in one case based on videoscopic observation after diagnostic navigation bronchoscopy but prior to tracer injection. Abbreviations: EBUS, endobronchial ultrasound; US, ultrasound; IQR, interquartile range; NB, navigation bronchoscopy; TBNA; transbronchial needle aspiration.

	**Characteristics**		**Frequency**
	Duration of NB, hh:mm, median (±IQR)		01:45 (±00:26)
	(Intraparenchymal) bleeding after biopsies, *n* (%) ^3^		11 (35.5%)
Pioneer Plus catheter	Tracer injection device, *n* (%)	Pioneer Plus catheter	30 (96.8%)
		Conventional TBNA needle	1 (3.2%)
	Tumor visibility on radial US imaging, *n* (%)		29 (96.7%)
	Real-time radial US visibility of tracer injection, *n* (%)		0 (0.0%)
	Multi-depot placement, *n* (%)	Yes (>1 injection)	22 (73.3%)
		No (1 injection)	8 (26.7%)
^99m^Tc-nanocolloid	Number of patients with injection(s), *n* (%)		31 (100%)
	Number of injections, median (±IQR)		2 (±1)
	Injection type, *n* (%)	Intratumoral	16 (51.6%)
		Peritumoral	7 (22.6%)
		Intra- and peritumoral	8 (25.8%)
	Injection volume, mL, median (±IQR)		0.43 (±0.5)
	Radioactivity, MBq, median (±IQR)		28.0 (±21.7)
	Total injection time per depot, seconds, median (±IQR)		9 (±9)
Iomeron 300	Number of patients with injection(s), *n* (%)		15 (48.4%)
	Number of injections, median (±IQR)		1 (±1)
	Injection type, *n* (%)	Intratumoral	11 (73.3%)
		Peritumoral	3 (20.0%)
		Intra- and peritumoral	1 (6.7%)
	Injection volume, mL, median (±IQR)		0.30 (±0.20)
	Injection visible on fluoroscopy, *n* (%)		15 (100%)
	Total injection time per depot, seconds, median (±IQR)		12 (±7.25)
	Leakage visible on fluoroscopy, *n* (%)		4 (26.7%)

**Table 3 cancers-16-03868-t003:** Post-navigation bronchoscopy parameters with scan times; SLN identification on SPECT imaging; pathology outcome of lesion and received treatment. ^4^ One patient had only a late scan-time and an extra late scan time at 23:51, which has been left out of the median and IQR calculations. ^5^ In one patient, an SLN could be identified on the early scan, but no late scan was performed due to logistical limitations. ^6^ based on SPECT imaging showing radioactivity throughout the ipsilateral lung. Abbreviations: AC, adenocarcinoma; CT, computed tomography; IQR, interquartile range; NB, navigation bronchoscopy; NSCLC, non-small cell lung cancer; SCC, squamous cell carcinoma; SPECT, single-photon emission computed tomography; SLN, sentinel lymph node.

	Characteristics		Frequency
SPECT/CT imaging	Patients with an early scan, *n* (%)		10 (32.3%)
	Early scan time, hh:mm, median (±IQR)		02:28 (±01:02)
	Patients with a late scan time, *n* (%)		30 (96.8%)
	Late scan time, hh:mm, median (±IQR) ^4^		04:19 (±00:55)
	SLN identification, all scan times, *n* (%)		10 (32.3%)
	SLN identification early scan, *n* (%)		5 (50.0%)
	SLN identification late scan, *n* (%) ^5^		9 (30.0%)
	Endobronchial leakage, *n* (%) ^6^		9 (29.0%)
Staging and treatment	Pathology outcome of lesion after NB, *n* (%)	AC	19 (61.3%)
		SCC	5 (16.1%)
		NSCLC	2 (6.5%)
		Benign	4 (12.9%)
		Non-representative	1 (3.2%)
	Treatment, *n* (%)	Surgery	15 (48.4%)
		Other	16 (51.6%)
	Subgroup: pathological lymph nodal stage after surgery, *n* (%)	pN0	12 (80.0%)
		pN0(i+)/isolated tumor cells	2 (13.3%)
		pN1	0 (0.0%)
		pN2	1 (6.7%)

## Data Availability

The data presented in this study are available in this article. Additional data can be shared up on reasonable request.

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
