# Peer review of "Feasibility of Non-Invasive Sentinel Lymph Node Identification in Early-Stage NSCLC Through Ultrasound Guided Intra-Tumoral Injection of 99mTc-Nanocolloid and Iodinated Contrast Agent During Navigation Bronchoscopy"

_cancers, 2024, doi:10.3390/cancers16223868_

Round 1

Reviewer 1 Report

Comments and Suggestions for Authors

The paper describes exhaustively not only the procedure, but also its limits, as low results in term of localization of sentinel nodes. It would be intriguing to know how the patients were stadiated before the procedure, with imaging techiniques as PET CT scan, but for the paucity of the population studied, this description would be futile as well.

Author Response

Comments 1:

The paper describes exhaustively not only the procedure, but also its limits, as low results in term of localization of sentinel nodes. It would be intriguing to know how the patients were stadiated before the procedure, with imaging techiniques as PET CT scan, but for the paucity of the population studied, this description would be futile as well

Response:

Thank you very much for taking the time to review this manuscript, your kind words and for asking a relevant question. It is important to stress that we have focused on a population referred for diagnosis of small peripheral lesions suspected for early stage lung cancer in general clinical stage I or state II at maximum. This means that in our population no patients with gross involvement of mediastinal nodes will be present since they will be diagnosed and staged by routine bronchoscopy and systematic EBUS. We have added the pre-procedural staging based on the available imaging in Table 1 and provided more detail on the population description in methods paragraph 2.1.

Reviewer 2 Report

Comments and Suggestions for Authors

Congratulations to you on your detailed analysis and comparison between methods of detecting SLN. The entire manuscript was well written with good language and layout except several typos which should be amended. 

Your results and conclusions are clear-cut showing the multi-modal catheter procedure was better than SPECT/CT imaging on the completeness of tumor visualization. That will be a contribution and a good reference to the future clinical practice.

One question : The cost of Pioneer plus ? And the comparison with other mentioned procedures.

Author Response

Comments 2:

Congratulations to you on your detailed analysis and comparison between methods of detecting SLN. The entire manuscript was well written with good language and layout except several typos which should be amended. 

Your results and conclusions are clear-cut showing the multi-modal catheter procedure was better than SPECT/CT imaging on the completeness of tumor visualization. That will be a contribution and a good reference to the future clinical practice.

One question : The cost of Pioneer plus ? And the comparison with other mentioned procedures.

Response:

Thank you very much for taking the time to review this manuscript. In the revised manuscript we have addressed all typo’s we’ve identified.

Regretfully we cannot answer your question regarding the costs of the Pioneer Plus for this use. We studied the feasibility of this device, outside it’s intended use, in this study. Furthermore pricing may differ in different regions and countries. Also the costs of the Boston Scientific iNod is unknown to us since it is not available outside of the USA.

Reviewer 3 Report

Comments and Suggestions for Authors

Congratulations to this elegant and important study. My questions are as follows.

1. What is the opinion of the authors, if the method is applicable after neoadjuvant treatment as well? 

2. Is there any difference in the success of the method in case of necrotic tumour compired to the non necrotic? 

Author Response

Comments 3:

Congratulations to this elegant and important study. My questions are as follows.

  1. What is the opinion of the authors, if the method is applicable after neoadjuvant treatment as well?
  2. Is there any difference in the success of the method in case of necrotic tumour compired to the non necrotic? 

Response:

Thank you very much for taking the time to review this manuscript. Please find detailed responses to your questions below and the corresponding revisions in track changes in the re-submitted files.

  1. While we believe this procedure could in theory be applicable in all patients, and developing technology that can help to evaluate the effect of neo-adjuvant treatment is indeed very important. We regretfully need to conclude that at this moment this technique is insufficiently sensitive to detect a possible effect of neo-adjuvant treatment on SLN involvement. In our study we excluded patients with previous local and systemic (lung) cancer treatments from participation. We have added a brief discussion on this interesting subject line 446-448.
  2. This is also a very interesting question and unfortunately again, not a subject that we can answer based on our results. Our study population are patients referred for navigation bronchoscopy usually have a small pulmonary nodule (median 18.7 mm) suspected of early stage lung cancer. None of these lesions had a necrotic component but in five cases peritumoral cysts could be identified. However, our study is not large enough to allow for in depth analysis of the effects of necrosis or cavities on the spread of 99mTc-labelled tracers. To answer your interesting question, larger studies are needed.